# Do community-based singing interventions have an impact on people living with dementia and their carers? A mixed-methods study protocol

Megan Polden  ,[1,2,3] Kerry Hanna,[4] Kym Ward,[5] Faraz Ahmed,[2] Heather Brown  ,[2] Carol Holland,[2,6] Hazel Barrow,[3] Jeanette Main,[6] Stella Mann,[3] Steve Pendrill,[6] Clarissa Giebel  [1,3]

¹Department of Primary Care & Mental Health, University of Liverpool, Liverpool, UK
²Division of Health Research, Lancaster University, Lancaster, UK
³NIHR Applied Research Collaboration North West Coast, Liverpool, UK
⁴School of Health Sciences, University of Liverpool, Liverpool, UK
⁵The Brain Charity, Liverpool, UK
⁶Lyrics and Lunch Charity, Lancaster, UK

**Correspondence to**
Megan Polden;
m.polden@liverpool.ac.uk

## ABSTRACT

**Introduction** Psychosocial interventions have been shown to improve mood, relieve stress and improve quality of life for people living with dementia (PwD). To date, most evaluations of singing interventions have focused on the benefits for PwD and not their carers. This research aims to evaluate the benefits of dementia singing groups for both PwD and their carers.

**Methods and analysis** This 2-year project will observe the impact of two different singing intervention services, one combining singing alongside dance and another that includes a sociable lunch. This project will aim to recruit a total of n=150 PwD and n=150 carers across the two singing interventions. Using a mixed-methods approach, the influence of both services will be analysed via the following outcome measures: quality of life, neuropsychiatric symptoms, social isolation, loneliness, cognition, carer burden and depressive symptoms in PwD and their carers using a prestudy/poststudy design. Regression models will be used to analyse the data with time (pre/post) as the exposure variable. Semistructured interviews will be conducted with a subset of people (n=40) to further investigate the impact of singing services with a specific focus on the acceptability of the interventions, barriers to access and prolonged engagement and potential for remote delivery. Interview data will be analysed using Braun and Clarke's reflexive thematic analysis, and public advisers will assist with coding the transcripts. A social return on investment analysis will be conducted to determine the social impact of the services.

**Ethics and dissemination** This project has received ethical approval from the University of Liverpool's Ethics Committee (App ref: 12374) and Lancaster University's Ethics Committee (App ref: 3442). All participants will provide informed consent to participate. Results will be presented at national and international conferences, published in scientific journals and publicly disseminated to key stakeholders.

## INTRODUCTION

Dementia currently affects more than 55 million people worldwide.[1] Dementia has detrimental effects on cognitive functioning[2]

**STRENGTHS AND LIMITATIONS OF THIS STUDY**

⇒ This study will assess the impact of two unique singing interventions for people living with dementia and their carers using a mixed methods approach and examine barriers and facilitators to accessing singing support services.
⇒ A social return on investment analysis will be conducted to determine the social return for each singing support service.
⇒ An important limitation of the study is that it does not include an independent control group.
⇒ The inability to follow-up after the 12-week intervention period means that the long-term effects of the interventions cannot be determined in this study.
⇒ As participants cannot be blinded, self-report measures may include an element of bias in the responses.

and often leads to behavioural and psychological changes, including aggression, anxiety, hallucinations and culturally inappropriate behaviours.[3–5] Dementia has profound effects on people's quality of life and ability to perform everyday tasks.[6–8] Pharmacological interventions can aid in the management of cognitive symptoms for PwD.[9] However, the benefits are often short-lived and are limited for behavioural symptoms such as anxiety, agitation and depressive symptoms.[10] Pharmacological approaches to manage behavioural symptoms in PwD include antipsychotics, anxiolytics, hypnotics and antidepressant medications.[11] There are often negative side effects associated with using these medications including increased occurrence of strokes[12] and mortality.[13 14] Research has also suggested that antipsychotics may worsen quality of life and cognition.[15–17] The negative side effects of pharmacological interventions highlight the need for non-pharmacological

interventions to reduce and help manage behavioural symptoms of dementia.[18]

As a result, there is a strong need for non-pharmacological interventions such as music therapy which has demonstrated strong benefits for PwD.[19 20] Music therapies have been shown to improve mood, regulate emotion and relieve stress in older adults[21] and in PwD.[22–24] Music therapy as a whole consists of many components and applications, and one specific and promising form of music therapy is singing interventions. The act of singing combines language, music and instinctive human behaviour to enhance neurological stimulation.[25] Research has linked singing to improved memory performance[26] and increased cognitive functioning as it utilises and engages brain pathways other than in plain speech, which is beneficial for people with advanced stages of dementia.[27 28] Group singing interventions may also help to improve social interactions between PwD, promoting relaxation and reducing levels of agitation.[29 30] Singing interventions in particular have demonstrated value and effectiveness within the realm of music therapies.

It should be noted that there is mixed evidence within the literature with a systematic review of music therapy for PwD finding that music listening had a greater effect on PwD compared with singing interventions.[19] However, in recent years, research has focused on specific aspects of music therapy which result in the best outcomes for PwD.[31–33] A systematic review of multiple variations of music therapy (ranging from passive music listening to active singing sessions) for PwD in residential care identified singing as an important aspect for change and improvements in mood and reductions in behavioural disturbances.[34] Further evidence from a randomised control trial suggests that singing interventions which actively engage participants may have greater effects on mood and quality of life when compared with passive music therapy.[31 35] These studies indicate that singing and active engagement in music sessions may be a key component for significant improvements in PwD. This variation in findings may be due to variations in dementia severity with some studies suggesting that passive music therapy may be more beneficial in advanced stages of dementia and active music therapy involving singing more beneficial in the early to moderate stages.[36]

PwD can often experience a decrease in physical activity levels post-diagnosis which can lead to adverse effects on physical health such as reductions in strength, balance, mobility and increased risk of physical frailty. Increased physical activity has been associated with positive cognitive and physical outcomes in PwD[37] and a reduction in symptom progression. Review evidence has found that multidomain interventions may be more beneficial for PwD.[38] Research examining dancing sessions in PwD has found that dancing can increase playfulness and sociability in PwD,[39] improve levels of agitation and cognitive function[40] and also improve physical performance such as balance and walking speed.[41] Interventions that combine physical aspects alongside cognitive stimulation

may lead to increased benefits and a wider range of positive outcomes. Research has not examined the effects of combining both singing and dancing.

Research has found that people with limited social networks and reduced social engagement may be more at risk of developing dementia compared with those with wider and richer social networks.[42] It is thought that social activity and engagement may be a protective factor against cognitive decline by increasing cognitive reserve to better maintain cognitive functioning and performance.[42] To our knowledge, no evaluations have examined the benefits of singing interventions combined with an eating-together element which provides a sociable experience alongside the singing session. Existing literature often focuses on the singing element of these interventions[43]; however, there are many social aspects of the sessions which lead to the benefits for PwD and their carers[44 45] that have previously been overlooked. The current project will investigate the impacts of two established charity-based services provided by The Brain Charity and the Lyrics and Lunch charity for PwD and their carers.

Most PwD live at home where they are often cared for and supported by their friends and family.[46] Family carers are a strong factor leading to more positive outcomes for PwD and can reduce or delay the need for residential care.[47] It can be hugely rewarding and fulfilling caring for a family member living with dementia[48]; however, it can also be challenging and have negative impacts on the carers' mental and physical health and quality of life.[47] Due to this, interventions such as music therapy have been developed to promote well-being among unpaid carers.[49] Singing interventions have been found to improve the relationship between PwD and their carers in addition to easing carer burden when attended together.[24] The challenges of dementia are often not limited to the PwD, with relatives, carers and friends often profoundly impacted.[50] Therefore, there is a need for services to offer support not only for PwD but also for their carers.[51 52]

Multiple barriers may prevent PwD from accessing and continuing to engage with an intervention such as cost of travel, increased reliance on paid and unpaid carers and cost of service and care.[53 54] Studies to date have not examined the potential barriers to accessing singing intervention services and specifically how these may differ based on socioeconomic position (SEP) and geographical location. Accessing services after a diagnosis can be difficult,[55] so understanding the benefits of a specific form of support and potential barriers to accessing these services is important. The current study examines barriers and facilitators for accessing and continuing to engage with singing intervention services and the organisations that provide these services.

Remote delivery of singing intervention services may be a solution to some barriers to attending singing interventions such as travel or reliance on carers. During the COVID-19 pandemic, multiple dementia support groups were moved online and demonstrated the potential for singing interventions to be delivered remotely.[33 56]

However, the feasibility, acceptability and whether remote delivery yields similar benefits as in-person groups have been questioned.[33] For people living with advanced stages of dementia, barriers may prevent them from attending in-person groups, and in these cases, remote delivery may be a method to improve access and reach of these services.

It is important to quantify the value of interventions, not just in economic terms but also in terms of social value. Social return on investment (SROI) is a method of cost-benefit analysis that assigns monetary values to social outcomes that usually would not be accounted for in standard financial evaluations. SROI analysis can be used to determine an intervention's social value in relation to financial investment.[57] SROI analysis can be a useful tool for examining interventions and to inform policy making relating to social support investments.[58 59] In the current project, the SROI will be examined for both support services.

This 2-year project aims to examine the benefits of established charity-provided dementia singing groups for both PwD and their unpaid carers using both quantitative and qualitative methods. For this, the following interlinked research questions will be addressed: (1) 'Do singing interventions combined with a social lunch have an impact on well-being in PwD and their carers'? (2) 'Do singing and dancing intervention services have an impact on well-being in PwD and their carers'? (3) 'What barriers are encountered when accessing singing intervention services and what prevents continued engagement'? (4) 'Could remote delivery of singing interventions improve access and what barriers may there be with remote delivery of singing interventions'? (5) 'What is the social return on investment of the singing support services examined in this project'?

## METHODS AND ANALYSIS
### Study overview
This mixed-methods study consists of four interlinked work packages (WPs) examining the influence of singing services provided by two registered third-sector organisations, Lyrics and Lunch and The Brain Charity. The barriers that people encounter when accessing and staying engaged with singing interventions will be examined and whether these disproportionately affect people based on sociodemographic factors specifically SEP and location (urban vs rural location). The potential of remote delivery of singing services to help improve wider access will also be assessed. To do this, semistructured interviews will be conducted with key stakeholders including PwD, unpaid carers, service providers (music therapists, workshop coordinators) and local facilitators (paid carers, care home managers). From this data, recommendations for future delivery of singing support services will be developed with a focus on improving access and continued engagement. An SROI will be conducted to examine the social impact of both support services. A systematic review is also being conducted on community-based singing interventions for people living with dementia and whether they improve mood and quality of life to further inform the work (Prospero ID: CRD42023395907).

### Two singing (and dancing) interventions
### Intervention 1: Lyrics and Lunch
Lyrics and Lunch is a charity based in Lancaster with the aim to serve people with dementia and their carers by providing a singing group and a nourishing lunch in a sociable community environment. The sessions take place weekly or biweekly and last approximately 2 hours with the first hour consisting of a sociable lunch followed by an hour of group singing.

For the singing session, attendees will be provided with a book with the lyrics to each song. Each session will start with the group singing an original song that allows each person to introduce themselves. The session leader will play or be accompanied by piano, guitar etc and facilitate the group to sing along. Familiar and well-known songs will be sung throughout the sessions such as 'Happy Together' by The Turtles. Around halfway through the sessions, percussion instruments will be given out and used to accompany around two to three of the songs. The session will often include rounds in which people will be split into smaller groups and each group will sing the same melody in tandem. Lyrics and Lunch is a non-denominational church-based organisation, and a short optional reflective spiritual component is included at the end, but the sessions are open to people with any religion or none. The service is open to PwD living within the community and their carers.

### Intervention 2: The Brain Charity's singing and dancing groups
The Brain Charity is based in Liverpool, and supports people with neurological conditions. The charity runs singing (Music Makes Us Sing) and dancing (Music Makes Us Dance) sessions in both community and residential care home settings for PwD and their carers. The singing sessions last approximately 1 hour and involve engaging in communal singing of familiar songs such as 'My Bonnie' by The Beatles led by two session leaders. Lyrics are presented on a TV screen, and attendees are positioned in a semicircle in view of the screen. The singing sessions have been designed alongside speech and language therapists and focus on language and breathing exercises to improve speech, pitch and rhythm such as call and response and clapping rhythm exercises. Attendees are encouraged to sing and dance along to the songs by session facilitators. The dance sessions last approximately 1 hour and involve physically engaging chair-based dance moves designed alongside a physical therapist and also incorporate group singing. The sessions are run by two session facilitators who before each song will demonstrate simple dance routines to familiar songs such as 'Hit the Road Jack' by Ray Charles. During the songs, session facilitators will give cues to the next dance moves so attendees can follow the dance routines. Props will be

used throughout the sessions such as scarves, percussion instruments and balloons. The sessions are run weekly in 12-week blocks and follow a consistent format each week using similar songs and dance routines so attendees can become familiar with them.

## Recruitment and sample size

For intervention 1 (Lyrics and Lunch): PwD and carers who have accessed the Lyrics and Lunch groups or who are beginning to attend will be invited to participate. A power analysis was conducted using G*Power software version 3.1.9.7. For the analysis, the power level was set at .80 with an error of .05.[60] The effect size (0.44) was based on a mean effect size taken across eight studies examining active music therapy and reported in Vasionytė et al[61] and based on the primary outcome measure of mood and depressive symptoms. Results revealed that a minimum sample size of n=55 is necessary to achieve a power of .80 at an alpha of .05. Therefore, to account for participant dropout and retention rates, the project will aim to recruit around n=75 PwD and n=75 of their carers from the Lyrics and Lunch groups.

For intervention 2 (The Brain Charity): PwD and carers who have expressed interest in accessing The Brain Charity's singing and dancing groups will be recruited before attending the groups. Separate samples will be collected for the singing intervention group and the dancing intervention group. Participants will be recruited from multiple singing and dancing intervention groups across the Liverpool area. These groups will include PwD living within the community and people living in residential care homes. The singing groups and dancing groups will be run by the same group of session facilitators.

The power analysis remained consistent with intervention 1 and revealed that a minimum sample size of n=55 is necessary to achieve a power of .80 at an alpha of .05. Therefore, across both intervention groups, a total sample size of n=300 will be recruited to account for participant dropout. This will consist of n=150 PwD (n=75 from the singing group and n=75 from the dancing group) and n=150 of their carers (n=75 from the singing group and n=75 from the dancing group). Data collection for the study commenced on the 25th of May 2023.

## Work package I: understanding the impacts of the Lyrics and Lunch support service on people with dementia and their carers

Work package I (WP1) will address the first research question by assessing the impact of a singing intervention service that includes a sociable lunch aspect on PwD and their carers. This study will use quantitative methods to examine the impact of attending the singing service on cognition, social isolation, loneliness, quality of life, neuropsychiatric symptoms such as agitation or other responsive behaviours, mood (depression/anxiety) and carer burden. Additionally, for the core outcome set of 13 outcomes of interventions valued by people living at home with dementia, developed directly with PwD, informal and professional carers[62] will be used. The 13 outcomes were classified into four domains including friendly neighbourhood and home, independence, self-managing dementia symptoms and quality of life and will be examined using a bespoke quantitative measure, an 18-item measure adapted from the Adults Social Care Outcome assessment,[63] the engagement and independence in dementia questionnaire[64] and Older Americans' Resources and Services Instrumental Activities of Daily Living assessment.[65] Outcome measures will be examined when the sessions are running (ON session) and compared with periods when the session is not running such as summer holidays and Christmas breaks (OFF session) (table 1). Participants will be required to have attended the Lyrics and Lunch sessions for a minimum period of 4 weeks before the ON session assessment. OFF session assessments will be completed after the sessions have not been attended for a minimum of 4 weeks. The Lyrics and Lunch sessions run weekly or biweekly.

## Measures

Standard outcome measures that will be assessed in PwD and their carers will include cognition assessed using a shortened version of Addenbrooke's cognitive examination (ACE-III)[66] where participant's verbal fluency and attention will be assessed, quality of life assessed using the dementia quality of life measure (DemQOL and DemQOL-Proxy),[67] social Isolation assessed using the Duke Social Support Index (DSSI-10),[68] loneliness assessed using three-item loneliness UCLA measure,[69] mood and depressive symptoms assessed using the Quick

**Table 1** Assessments to be completed by patients living with dementia and their carers

| Assessment time period | Assessment completed by person living with dementia | Assessment completed by unpaid carer |
|---|---|---|
| ON session (participant has attended service for a minimum of 4 weeks) | QIDS-SR, ACE, DemQOL, DSSI-10 | QIDS-SR, DSSI-10, NPI-Q, DemQOL-Proxy, Zarit burden questionnaire |
| OFF session (participant has not attended a session for a minimum of 4 weeks) | QIDS-SR, ACE, DemQOL, DSSI-10 | QIDS-SR, DSSI-10, NPI-Q, DemQOL-Proxy, Zarit burden questionnaire |

ACE, Addenbrooke's cognitive examination; DemQOL, dementia quality of life; DSSI-10, Duke Social Support Index; NPI-Q, Neuropsychiatric Inventory Questionnaire; QIDS-SR, Quick Inventory of Depressive Symptomatology.

Inventory of Depressive Symptomatology (QIDS-SR),[70] neuropsychiatric symptoms assessed using Neuropsychiatric Inventory Questionnaire (NPI-Q) completed by carers[71] and carer burden assessed using the Zarit burden questionnaire.[72]

Demographic information will be collected from participants at the first assessment including age, gender, ethnicity, highest education level and number of years in formal education (an indicator of SEP), years since diagnosis and years since symptom onset. Participant attendance will be recorded to determine adherence to the intervention. The Global Deterioration Scale for Assessment of Primary Degenerative Dementia will be completed by carers to determine dementia severity.[73]

It is hypothesised that attendance to singing interventions will increase quality of life and mood and reduce depressive and neuropsychiatric symptoms in PwD and their carers. Social isolation can be prominent in PwD, and attending the singing intervention may reduce feelings of social isolation. Due to this, it is hypothesised that attending the singing intervention will result in reduced self-reported feelings of social isolation and loneliness in both PwD and their carers. It is hypothesised that both PwD and their carers will show improvements in the core outcome measures during the times that they are attending the singing intervention compared with the period when they are not attending the sessions.

### Data analysis

To examine whether there were changes in outcome variables during on and off periods and to assess whether singing intervention support services are associated with changes in mood/depressive symptoms, quality of life, cognition and social isolation in people living with dementia and their carers, linear regression models will be used with time (on vs off) as the exposure variable. For the model examining outcomes in people living with dementia, outcome variables will be the QIDS-SR score, ACE-III score, DemQOL score, DSSI-10 score, NPI-Q score and core outcome set score. Covariates will include participants' age, years since diagnosis, years since symptom onset and number of sessions attended. A separate model will examine outcome measures in dementia carers, and outcome measures will be DSSI-10 score, QIDS-SR score and carer burden.

### Work package II: understanding the impacts of singing and dancing support service conducted by The Brain Charity on people with dementia and their carers

WP2 will address the second research question and the impacts of attending either a 12-week singing intervention or a 12-week dancing intervention that also involves some singing for PwD and their carers. Participants will only attend one intervention service (either the singing or the dancing intervention). The benefits of dementia singing and dancing groups for both PwD and their unpaid carers, both community residing and those living in care homes (conducted within the care home), will

be examined using a mixed-methods approach. The project will assess the influence of the services on cognition, mood, social isolation, loneliness, carer burden and neuropsychiatric symptoms.

### Measures

The following outcome measures will be used: DemQOL, DemQOL-Proxy, QIDS-SR, ACE-III (shortened version assessing verbal fluency and attention only), NPI-Q, DSSI-10, and Zarit burden questionnaire (figure 1). Outcome measures will be examined at baseline (before the start of the 12-week block of sessions) and then 1 week after the final session, thus providing preintervention and postintervention measures. The QIDS-SR will be completed every 4 weeks during the 12 weeks to examine more subtle changes in mood and depressive symptoms over the 12 weeks (figure 1). The dance elements included in the dancing sessions may have impacts on the physical health and frailty of PwD attending the sessions. Due to this, the physical frailty phenotype[74] measure of physical frailty will be included for the dancing group that will assess weight loss, grip strength, exhaustion, gait speed and physical activity. The physical frailty phenotype will be completed preintervention and postintervention.

Similar to WP1, participant attendance will be recorded to determine adherence to the intervention. Demographic information will be collected at baseline, including age, gender, ethnicity, highest education level (an indicator of SEP), number of years in formal education (an indicator of SEP), years since diagnosis and years since symptom onset. The Global Deterioration Scale for Assessment of Primary Degenerative Dementia will be completed by carers to determine dementia severity.[73]

### Data analysis

Consistent with WP1, we will examine whether there were changes in outcome variables preintervention versus postintervention and assess whether singing and dancing interventions are associated with changes in mood/depressive symptoms, quality of life, cognition and social isolation in people living with dementia and their carers. Linear regression models will be used with time (pre/post) and intervention type (singing or dancing intervention) as the exposure variables. The outcome variables will be QIDS-SR score, ACE-III score, DemQOL score, DSSI-10 score, Zarit burden score, NPI-Q score, sit-to-stand assessment score and gait assessment score. Covariates will include participants' age, years since diagnosis, years since symptom onset, ACE score (dementia severity), number of sessions attended and participant residence (community dwelling or care home). Separate models will be conducted to examine the singing and dancing service's effect on people living with dementia and then their carers and for each of the outcome measures.

 

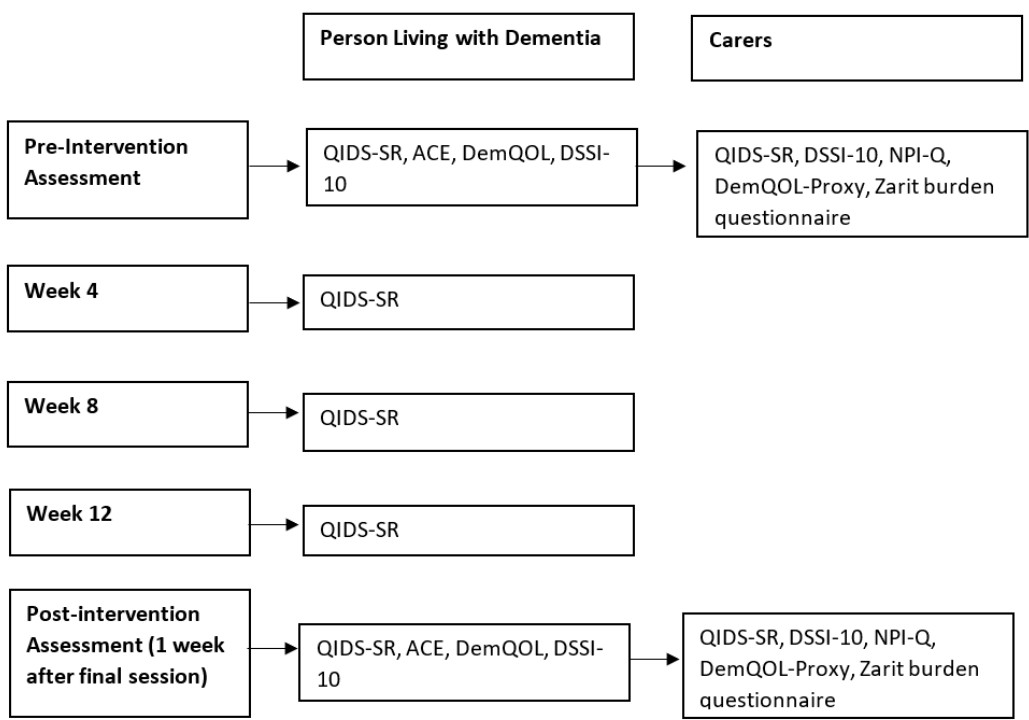

**Figure 1** Study timeline. ACE, Addenbrooke's cognitive examination; DemQOL, dementia quality of life; DSSI-10, Duke Social Support Index; NPI-Q, Neuropsychiatric Inventory Questionnaire; QIDS-SR, Quick Inventory of Depressive Symptomatology.

### Work package III: understanding the acceptability, remote delivery and barriers to accessing and continued engagement with singing interventions

WP3 will address research questions 3 and 4 and will examine barriers to accessing and continued engagement with singing interventions and how these factors may differ based on SEP and location (urban vs rural location). Accessibility barriers are likely to vary across these areas and depend on people's SEP and should be examined and acknowledged when rolling out intervention programmes on a wider scale.

A series of semistructured follow-up interviews will be conducted with a subset of PwD, unpaid carers, service providers (music therapists and workshop coordinators) and local facilitators (paid carers and care home managers). The topic guide(s) will be coproduced with the teams and four public advisers consisting of a person living with dementia, a paid carer, an unpaid carer and a music teacher (JM) who runs the sessions.

The first aim will focus on barriers to access and continued engagement with the intervention service. To address this, a subset of 20 people will be recruited (approx. 10 PwD and 10 unpaid carers) from both interventions, resulting in a total of 40 interviews. This initial sample size determination was guided by information power methods.[75] These interviews will focus on the following topics: acceptability of the services, impacts of the service on mood and overall quality of life and accessibility including barriers to access and continued engagement.

Our second aim is to examine the potential of services to be adapted for remote delivery and potential barriers. When discussing remote access, adaptations made to the services when COVID-19 restrictions were in place will be discussed and how successful these remote access adaptations were. To address this, approximately 20 people will be recruited: 10 service providers (music therapists and workshop coordinators) and 10 local facilitators (paid carers and care home managers). These interviews will focus on the following topics: impacts of the service on PwD and their carers and barriers to remote delivery of singing. Again, this initial sample size determination was guided by information power methods.[75] Preliminary analysis will be used to determine the power of the analysis during the data collection phase to determine if the sample size calculation is correct. After the first three interviews, an initial review of the data will be conducted, and initial suggestions of relevant theory/themes will be made. This assessment will be considered again before closing data collection. Data will be analysed using Braun and Clarke's reflexive thematic analysis,[76] and public advisers will assist with coding the transcripts.

### Work package IV: examination of the social impact of singing interventions

WP4 will address research question 5 and the social impact of singing interventions examined by conducting an evaluative SROI analysis on both services. Using standardised methods and following the seven principles, an SROI will be conducted.[77] Separate SROI analyses will be conducted

for the Lyrics and Lunch charity and The Brain Charity. The methods and procedure employed for both analyses will remain consistent. An impact map will be created to calculate the social value generated by the intervention service which involves six stages.[78] Stage 1 will identify key stakeholders and people likely to be impacted by the services such as PwD, relatives, unpaid and paid carers, Lyrics and Lunch and The Brain Charity. Stage 2 will use quantitative data collected for WPs I and II on the effects of the intervention on quality of life, mood, carer burden, depressive symptoms and agitation levels and will map identified outcomes creating an impact map. This stage will identify what changes occurred and for whom. Stage 3 will provide a monetised value to each of the outcomes including those that do not have a price attached to them such as changes in quality of life. For outcome measures that do not have a clear monetary value, established SROI value banks such as Social Value UK will be used to identify suitable financial proxies.[79] Stage 4 will establish impact by accounting for attribution, deadweight, displacement and drop-off. These adjustments are made to ensure that social value is not overclaimed.[77] Percentage values will be determined for attribution which considers changes or outcomes declared in the analysis that may not be due to just the singing intervention but may be due to other support services. A percentage value for the deadweight will be determined which considers how much change in the outcome values would have happened regardless of the singing intervention programme. Deadweight will be calculated as a 10% reduction in overall value for every resource the participant identifies as an alternative to the singing support service, for example, if participants attend other support groups. Stage 5 will calculate the SROI benefits by adding up and subtracting any negatives (attribution, distribution, drop-off and deadweight), and this will be compared with the investment cost of the service. This calculation will provide us with an SROI ratio which demonstrates the social value of the service in relation to the cost invested, for example, an SROI ratio of £3.50: £1 means that for every pound invested, there is £3.50 of social value created. Stage 6 will disseminate the findings with key stakeholders and recommendations created for future delivery of the service.

### Patient and public involvement

This project has been developed and conceptualised alongside members of the public and key stakeholders. Our formal project team includes four public advisers including a person living with dementia (SM), two carers (one paid (SP) and one unpaid (HB)) and a music teacher (JM) who runs the singing sessions. All public advisers have previous experience and involvement with the singing sessions provided by The Brain Charity or the Lyrics and Lunch charity. Public advisers contribute to all aspects of this research project, from designing study documents and advising on outcome measures to the interpretation of findings and dissemination of the research. Public advisers are reimbursed for the time they contribute to the project.

## ETHICS AND DISSEMINATION

This project has received ethical approval from the University of Liverpool's Ethics Committee (App ref: 12374) and Lancaster University's Ethics Committee (App ref: 3442). All methods will be conducted in accordance with the relevant guidelines and regulations, and all participants will provide informed consent to participate. Findings from this project will be presented at national and international conferences and published in scientific journals. Findings will be publicly disseminated to key stakeholders and within the community, and the organisations included in this study will use the outcomes of the research for their own continuous improvement strategies.

## DISCUSSION

Using a mixed-methods design, this project seeks to explore the influence of two established singing intervention services for PwD and their carers. The study aims to collect a large sample of key stakeholders to assess the influence of attending singing support services on well-being and quality of life through a combination of quantitative and qualitative methods. The findings from this research will highlight the impact of singing support services for PwD and their carers and will allow the SROI of such services to be evaluated. The project will examine the barriers that people encounter when accessing and continuing to engage with singing interventions and whether these barriers disproportionately affect people from lower SEP. It can be challenging to access support services after a diagnosis of dementia,[55] and exploring ways to make services more widely and easily accessible is important. Methods to reduce potential barriers to access such as travel, increased reliance on carers and availability of services will be explored alongside the potential for remote delivery of singing services to improve and expand access.

A limitation of this study that should be noted is the lack of an independent control group. This study is therefore unable to robustly examine the impact of each intervention. However, it will provide important information on the acceptability and potential impact of the support services for PwD and their carers and barriers to access or engagement. This study will provide an important knowledge base for future studies and evidence on whether larger randomised control trials are warranted. A further limitation is that as participants cannot be blinded, self-report measures may include an element of bias in the responses. Additionally, this study is unable to determine the long-term effects of the interventions and cannot determine whether any effects of the intervention persist or diminish after the 12-week intervention. Future

research should include further follow-up assessments postintervention to determine long-term effects.

If the results from the current study suggest that singing support services have a positive impact on PwD and their carers, then support services could be scaled up and applied to other regions and settings to improve access to beneficial support services. Examining the SROI will provide insight into the cost-effectiveness of the support services and if they will be suitable to apply in lower-income areas and regions both nationally and internationally. The findings from this project will be used to develop recommendations for future delivery of singing intervention services.

**Acknowledgements** We would like to thank The Brain Charity and Lyrics and Lunch Charity for their involvement in this research.

**Contributors** Conceptualisation and methodology design contributions and project administration contributions were made by all authors: MP, KH, KW, FA, HB, CH, HB, JM, SM, SP and CG. The original draft preparation contributions were made by MP. MP, KH, KW, FA, HB and CH contributed to revising the manuscript. All authors MP, KH, KW, FA, HB, CH, HB, JM, SM, SP and CG read and agreed to the published version of the manuscript.

**Funding** National Institute for Health and Care Research Applied Research Collaboration North West Coast (ARC NWC) (Award/Grant no: NA) and The Alzheimer's Society (Award/Grant no: NA).

**Competing interests** Carol Holland, Jeanette Main and Steve Pendrill are trustees of the Lyrics and Lunch charity. All other authors declare no conflict of interest.

**Patient and public involvement** Patients and/or the public were involved in the design, or conduct, or reporting, or dissemination plans of this research. Refer to the Methods section for further details.

**Patient consent for publication** Not applicable.

**Provenance and peer review** Not commissioned; externally peer reviewed.

**ORCID iDs**
Megan Polden http://orcid.org/0000-0002-1813-0765
Heather Brown http://orcid.org/0000-0002-0067-991X
Clarissa Giebel http://orcid.org/0000-0002-0746-0566

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
