## [Reviewer comments · BMJ Open]

ARTICLE DETAILS

TITLE (PROVISIONAL)	Do community-based singing interventions have an impact on people living with dementia and their carers? A mixed-methods study protocol
AUTHORS	Polden, Megan; Hanna, Kerry; Ward, Kym; Ahmed, Faraz; Brown, Heather; Holland, Carol; Barrow, Hazel; Main, Jeanette; Mann, Stella; Pendrill, Steve; Giebel, Clarissa

VERSION 1 – REVIEW

REVIEWER	McCleery, Jenny Oxford Health NHS Foundation Trust, Mental Health
REVIEW RETURNED	25-Jul-2023

GENERAL COMMENTS	This is a protocol for a worthwhile pragmatic study. It has potential to produce useful results to inform third sector projects for people with dementia and their carers. I understand that the study is already underway so have only a few comments for the authors. An effect size from the literature was used for the power calculations, but an effect size for what outcome(s)? The manuscript would benefit from more information about the analyses planned for the quantitative (scale) measures. Without this, the study is not replicable. I am not qualified to comment on the SROI analysis plan. I liked the ON versus OFF design for WP1, but for WP2 it appeared that outcomes would be collected no later than 1 week after the end of the intervention. It would be very helpful to have later data collection to determine whether any detectable effects persisted.
--

REVIEWER	Amano, Takashi Rutgers University Newark
REVIEW RETURNED	02-Sep-2023

GENERAL COMMENTS	Thank you to the authors for submitting their manuscript describing the protocol for evaluating singing intervention services for people with dementia and their carers. While the manuscript holds potential for publication, I have identified several concerns that need to be addressed. Please refer to my specific comments below. BACKGROUND  In the second paragraph of the introduction section, the authors discuss studies of "music therapy and singing interventions." I
---

suggest that they clarify the distinction between these two approaches and focus on presenting evidence specific to singing interventions. The flow of this section could be improved as follows: (1) Non-pharmacological interventions show promise, (2) Music therapy is among the most promising, and (3) Singing interventions have demonstrated effectiveness and value within the realm of music therapies.

- On page 5, the authors discuss the benefits of combining physical and social components with singing interventions. It is advisable to provide a theoretical rationale for this choice. While it is understood that singing, dancing, and social aspects of activities can be beneficial, it is not clear why these particular components were selected. For instance, why were cognitive activities not considered for inclusion? Are the authors expecting to observe synergistic effects by combining these elements? Additionally, it is not well explained why only two combinations, "singing and social lunch" or "singing and dancing," were chosen. Furthermore, the meaning of "benefits" is not clearly articulated. Singing, dancing, and socialization may each offer distinct advantages, so it would be helpful to clarify whether the authors are attempting to address the limitations of singing interventions through these combinations. If so, they should explicitly discuss their specific objectives in this section, as the aims presented later are quite broad. If not, please clarify what specific outcomes the authors expect to observe by combining these three activities.

- Specific aims 3, 4, and 5 are not thoroughly explained in the introduction.

- o Specific aim 3: While the authors mention barriers to accessing formal services for people with dementia, it is not evident why the focus should specifically be on barriers to singing interventions. In this study, singing interventions are provided as a service by organizations that the participants are already connected to. It might be more relevant to discuss the accessibility of these organizations rather than the specific interventions. Please expand upon this point to provide greater clarity.

- o Specific aim 4: The concept of "remote delivery" is introduced without prior discussion in the introduction. It would be beneficial to include a literature review on remote delivery to provide context and rationale for this aim.

- o Specific aim 5: The term "social return" is not clearly defined and remains unexplored in the introduction. Please provide a clear definition and discuss its relevance in the context of this study.

METHODS

- The descriptions of the two interventions are too brief. For instance, "The second hour involves group singing of familiar songs and includes percussion instruments with some of the songs." This is the only description provided regarding the singing intervention at Lyrics and Lunch. I recommend that the authors provide more detailed descriptions of the interventions at both sites to ensure clarity and comprehensiveness.

- Please specify the statistical analyses that will be conducted to examine the effectiveness of singing interventions. Given that this study is not a randomized controlled trial (RCT), it is particularly important to address how the authors plan to account for confounding factors and potential biases in their analyses.

- As previously mentioned, the explanation regarding "social return" is insufficient. Since I am not familiar with social return on investment (SROI) analysis, I suggest that the authors provide references for this analysis and demonstrate that they are utilizing a

VERSION 1 – AUTHOR RESPONSE

Reviewer 1 Comments:

Comment 1: An effect size from the literature was used for the power calculations, but an effect size for what outcome(s)?

Response: The power analysis was based on the primary outcome measure of mood and depressive symptoms, this information has been added to the manuscript on page 9.

Comment 2: The manuscript would benefit from more information about the analyses planned for the quantitative (scale) measures. Without this, the study is not replicable.

Response: Thank you for this comment. We have added a data analysis section detailing the planned quantitative analysis. The following sections have been added to the manuscript on page 11 and 12.

“2.4.2. Data Analysis

To examine whether there were changes in outcome variables during on and off periods and to assess whether singing intervention support services are associated with changes in mood/depressive symptoms, quality of life, cognition and social isolation in people living with dementia and their carers, linear regression models will be used with time (on vs off) as the exposure variable. For the model examining outcomes in people living with dementia, outcome variables will be QIDS-SR score, ACE-III score, DemQOL score, DSSI-10 score, NPI-Q score and Core outcome set score. Covariates will include participant age, years since diagnosis, years since symptom onset and number of sessions attended. A separate model will examine outcome measures in dementia carers, outcome measures will be DSSI-10 score, QIDS-SR score and carer burden.

“2.5.2 Data Analysis

Consistent with WP1, we will examine whether there were changes in outcome variables pre vs. post-intervention and assess whether singing and dancing interventions are associated with changes in mood/depressive symptoms, quality of life, cognition and social isolation in people living with dementia and their carers. Linear regression models will be used with time (pre/post) and intervention type (singing or dancing intervention) as the exposure variables. The outcome variables will be QIDS-SR score, ACE-III score, DemQOL score, DSSI-10 score, Zarit burden score, NPI-Q score, sit-to-stand assessment score and gait assessment score. Covariates will include participant age, years since diagnosis, years since symptom onset, ACE score (dementia severity) and number of sessions attended. Separate models will be conducted to examine the singing and dancing services effect on people living with dementia and then their carers and for each of the outcome measures.”

Comment 3: I liked the ON versus OFF design for WP1, but for WP2 it appeared that outcomes would be collected no later than 1 week after the end of the intervention. It would be very helpful to have later data collection to determine whether any detectable effects persisted.

Reviewer 2 Comments: Unfortunately, is not within the funding scope for this project. We acknowledge that this is a limitation of the study and future research future research would benefit from to examining the longevity of any detectable effects.

BACKGROUND

Comment 1: In the second paragraph of the introduction section, the authors discuss studies of "music therapy and singing interventions." I suggest that they clarify the distinction between these two approaches and focus on presenting evidence specific to singing interventions. The flow of this section could be improved as follows: (1) Non-pharmacological interventions show promise, (2) Music

therapy is among the most promising, and (3) Singing interventions have demonstrated effectiveness and value within the realm of music therapies.

Response: Thank you for this comment, we have revised this section to improve the flow and also clarity of the introduction. The revised paragraph is presented below and on page 4.

“As a result, there is a strong need for non-pharmacological interventions such as music therapy which has demonstrated strong benefits for PwD¹⁹⁻²⁰. Music therapies have been shown to improve mood, regulate emotion and relieve stress in older adults²¹ and in PwD²²⁻²⁴. Music therapy as a whole consists of many components and applications and one specific and promising form of music therapy is singing interventions. The act of singing combines language, music and instinctive human behaviour to enhance neurological stimulation²⁵. Research has linked singing to improved memory performance²⁶ and increased cognitive functioning as it utilises and engages brain pathways other than in plain speech, which is beneficial for people with advanced stages of dementia^{27,28}. Group singing interventions may also help to improve social interactions between PwD, promoting relaxation and reducing levels of agitation^{29,30}. Singing interventions in particular have demonstrated value and effectiveness within the realm of music therapies.”

Comment 2: On page 5, the authors discuss the benefits of combining physical and social components with singing interventions. It is advisable to provide a theoretical rationale for this choice. While it is understood that singing, dancing, and social aspects of activities can be beneficial, it is not clear why these particular components were selected. For instance, why were cognitive activities not considered for inclusion? Are the authors expecting to observe synergistic effects by combining these elements? Additionally, it is not well explained why only two combinations, "singing and social lunch" or "singing and dancing," were chosen. Furthermore, the meaning of "benefits" is not clearly articulated. Singing, dancing, and socialization may each offer distinct advantages, so it would be helpful to clarify whether the authors are attempting to address the limitations of singing interventions through these combinations. If so, they should explicitly discuss their specific objectives in this section, as the aims presented later are quite broad. If not, please clarify what specific outcomes the authors expect to observe by combining these three activities.

Response: Thank you for this comment. The following paragraph has been added to provide more explanation and justification for the choice to examine singing and dancing collectively. Further justification has also been added to provide more information and reasoning behind the choice to examine an intervention that combines a lunch and singing aspect. We have also specified what specific benefits we predict each intervention may result in. The following sections have been expanded and revised in the manuscript on page 5.

“PwD can often experience a decrease in physical activity levels post diagnosis which can lead to adverse effects on physical health such as reductions in strength, balance, mobility and increased risk of physical frailty. Increased physical activity has been associated with positive cognitive and physical outcomes in PwD³⁷ and a reduction in symptom progression. Review evidence has found that multidomain interventions may be more beneficial for PwD³⁸. Research examining dancing sessions in PwD has found that dancing can increase playfulness and sociability in PwD³⁹, improve levels of agitation and cognitive function⁴⁰ and also improve physical performance such as balance and walking speed⁴¹. Interventions that combine physical aspects alongside cognitive stimulation may lead to increased benefits and a wider range of positive outcomes. Research has not examined the effects of combining both singing and dancing. It is possible that an intervention combining both physical (dancing) and cognitive (singing) elements could lead to increased benefits for PwD and their carers.

Research has found that people with limited social networks and reduced social engagement may be more at risk of developing dementia compared to those with wider and richer social networks⁴². It is thought that social activity and engagement may be a protective factor against cognitive decline by increasing cognitive reserve to better maintain cognitive functioning and performance⁴². To our knowledge, no evaluations have examined the benefits of singing interventions combined with an eating-together element which provides a sociable experience alongside the singing session. Existing literature often attributes the benefits from singing interventions directly to the singing element⁴³, however it may be the social aspects of the sessions which lead to the benefits for PwD and their

carers^{44,45}. The current project will investigate the impacts of two established charity-based services provided by The Brain Charity and the Lyrics and Lunch charity for PwD and their carers.”

Comment 3: Specific aims 3, 4, and 5 are not thoroughly explained in the introduction.

Response: We have revised the manuscript and added further information relating to these aims on page 6. The following sections have been revised and added to the manuscript.

“There are multiple barriers that may prevent PwD from accessing and continuing to engage with an intervention such as cost of travel, increased reliance on paid and unpaid carers and cost of service and care^{54,55}. Studies to date have not examined the potential barriers to accessing singing intervention services and specifically how these may differ based on social-economic position and geographical location. Accessing services after a diagnosis can be difficult⁵⁶ so understanding the benefits of a specific form of support and potential barriers to accessing these services is important. The current study examines barriers and facilitators for accessing and continuing to engage with singing intervention services and the organisations that provide these services.

Remote delivery of singing interventions services may be a solution to some barriers for attending singing interventions such as travel or reliance on carers. During the COVID-19 pandemic multiple dementia support groups were moved online and demonstrated the potential for singing interventions to be delivered remotely^{57,58}. However, the feasibility, acceptability and whether remote delivery yields similar benefits as in-person groups has been questioned⁵⁷. For people living with advanced stages of dementia barriers may prevent them attending in-person groups and in these cases remote delivery may be a method to improve access and reach of these services.

It is important to quantify the value of interventions, not just in economic terms, but also in terms of social value. Social Return on Investment (SROI) is a method of cost-benefit analysis that assigns monetary values to social outcomes that usually would not be accounted for in standard financial evaluations. SROI analysis can be used to determine an intervention’s social value in relation to financial investment⁵⁹. SROI analysis can be a useful tool for examining interventions and to inform policy-making relating to social support investments^{60,61}. In the current project, the SROI will be examined for both support services.”

Comment 4: Specific aim 3: While the authors mention barriers to accessing formal services for people with dementia, it is not evident why the focus should specifically be on barriers to singing interventions. In this study, singing interventions are provided as a service by organizations that the participants are already connected to. It might be more relevant to discuss the accessibility of these organizations rather than the specific interventions. Please expand upon this point to provide greater clarity.

Response: Thank you for this comment, in a number of cases people were not previously connected with the organisation however, we have reworded this to improve clarity and state that we will also focus on facilitators as well as barriers. We will examine access and engagement with singing interventions services and this will also include questions relating to how PwD first heard and accessed the service via the organisation that provides it. The section below has been added to the manuscript on page 6 to provide more information on this.

“The current study examines barriers and facilitators for accessing and continuing to engage with singing intervention services and the organisations that provide these services.”

Comment 5: Specific aim 4: The concept of "remote delivery" is introduced without prior discussion in the introduction. It would be beneficial to include a literature review on remote delivery to provide context and rationale for this aim.

Response: Thank you for this comment, we have added discussion of this topic to the introduction section of the manuscript. The following paragraph has been added to the manuscript on page 6.

“Remote delivery of singing interventions services may be a solution to some barriers for attending singing interventions such as travel or reliance on carers. During the COVID-19 pandemic multiple dementia support groups were moved online and demonstrated the potential for singing interventions

to be delivered remotely^{57,58}. However, the feasibility, acceptability and whether remote delivery yields similar benefits as in-person groups has been questioned⁵⁷. For people living with advanced stages of dementia barriers may prevent them attending in-person groups and in these cases remote delivery may be a method to improve access and reach of these services.”

Comment 6: Specific aim 5: The term "social return" is not clearly defined and remains unexplored in the introduction. Please provide a clear definition and discuss its relevance in the context of this study.

Response: Thank you for this comment, we have added a definition to the introduction and provided more information on the relevance of this method. The paragraph below has been added to the manuscript on page 6.

“It is important to quantify the value of interventions, not just in economic terms, but also in terms of social value. Social Return on Investment (SROI) is a method of cost-benefit analysis that assigns monetary values to social outcomes that usually would not be accounted for in standard financial evaluations. SROI analysis can be used to determine an intervention’s social value in relation to financial investment⁵⁹. SROI analysis can be a useful tool for examining interventions and to inform policy-making relating to social support investments^{60,61}. In the current project, the SROI will be examined for both support services.”

METHODS

Comment 7: The descriptions of the two interventions are too brief. For instance, "The second hour involves group singing of familiar songs and includes percussion instruments with some of the songs." This is the only description provided regarding the singing intervention at Lyrics and Lunch. I recommend that the authors provide more detailed descriptions of the interventions at both sites to ensure clarity and comprehensiveness.

Response: Thank you for this comment, we have added further details to the descriptions of the two interventions. The following sections have been added to the manuscript on page 8.

“The Brain Charity is based in Liverpool, and they support people with neurological conditions. The charity runs singing (Music Makes Us Sing) and dancing (Music Makes Us Dance) sessions in both community and residential care home settings for PwD and their carers. The singing sessions last approximately one hour and involve engaging in communal singing of familiar songs such as “My Bonnie” by The Beatles lead by two session leaders. Lyrics are presented on a TV screen and attendees are positioned in a semi-circle in view of the screen. The singing sessions have been designed alongside speech and language therapists and focus on speech, language and breathing exercises to improve speech, pitch and rhythm such as call and response and clapping rhythm exercises. Attendees are encouraged to sing and dance along to the songs by session facilitators. The dance sessions last approximately one hour and involve physically engaging chair-based dance moves designed alongside a physical therapist and also incorporates group singing. The sessions are run by two session facilitators who prior to each song will demonstrate simple dance routines to familiar songs such as “Hit the road jack” by Ray Charles. During the song’s, session facilitators will give cues to the next dance moves so attendees are able to follow the dance routines. Props will be used throughout the sessions such as scarves, percussion instruments and balloons. The sessions are run weekly in 12-week blocks and follow a consistent format each week using similar songs and dance routines so attendees are able to become familiar with them.”

“The sessions take place weekly or biweekly and last approximately 2 hours with the first hour consisting of a sociable lunch followed by an hour of group singing. For the singing session attendees will be provided with a book with the lyrics to each song. Each session will start with the group singing an original song that allows each person to introduce themselves. The session leader will play piano or guitar and facilitate the group to sing along. Familiar and well-known songs will be sung throughout the sessions such as “Happy together” by The Turtles. Around half way through the sessions, percussion instruments will be given out and used to accompany the around 2-3 songs. The session will often include rounds in which people will be split into smaller groups and each group will sing the same melody in tandem. Lyrics and Lunch is a non-denominational Church-based organisation and a short optional reflective spiritual component is included at the end, but the sessions are open to

people with any religion or none. The service is open to PwD living within the community and their carers.”

Comment 8: Please specify the statistical analyses that will be conducted to examine the effectiveness of singing interventions. Given that this study is not a randomized controlled trial (RCT), it is particularly important to address how the authors plan to account for confounding factors and potential biases in their analyses.

Response: Details on the planned statistical analysis have been added to the methods section. The following sections have been added to describe the planned quantitative analysis on page 11 and 12.

“2.4.2. Data Analysis

To examine whether there were changes in outcome variables during on and off periods and to assess whether singing intervention support services are associated with changes in mood/depressive symptoms, quality of life, cognition and social isolation in people living with dementia and their carers, linear regression models will be used with time (on vs off) as the exposure variable. For the model examining outcomes in people living with dementia, outcome variables will be QIDS-SR score, ACE-III score, DemQOL score, DSSI-10 score, NPI-Q score and Core outcome set score. Covariates will include participant age, years since diagnosis, years since symptom onset and number of sessions attended. A separate model will examine outcome measures in dementia carers, outcome measures will be DSSI-10 score, QIDS-SR score and carer burden.

“2.5.2 Data Analysis

Consistent with WP1, we will examine whether there were changes in outcome variables pre vs. post-intervention and assess whether singing and dancing interventions are associated with changes in mood/depressive symptoms, quality of life, cognition and social isolation in people living with dementia and their carers. Linear regression models will be used with time (pre/post) and intervention type (singing or dancing intervention) as the exposure variables. The outcome variables will be QIDS-SR score, ACE-III score, DemQOL score, DSSI-10 score, Zarit burden score, NPI-Q score, sit-to-stand assessment score and gait assessment score. Covariates will include participant age, years since diagnosis, years since symptom onset, ACE score (dementia severity) and number of sessions attended. Separate models will be conducted to examine the singing and dancing services effect on people living with dementia and then their carers and for each of the outcome measures.”

Comment 9: As previously mentioned, the explanation regarding "social return" is insufficient. Since I am not familiar with social return on investment (SROI) analysis, I suggest that the authors provide references for this analysis and demonstrate that they are utilizing a standard methodology.

Response: This section has been revised to add more information and further referencing to highlight that the methods used is a standardised method. The following section has been revised and extended on page 14.

“Using standardised methods and following the seven principles a SROI will be conducted⁷⁹. Separate SROI analyses will be conducted for the Lyrics and Lunch Charity and The Brain Charity. The methods and procedure employed for both analyses will remain consistent. An impact map will be created to calculate the social value generated by the intervention service which involves six stages⁸⁰. Stage 1 will identify key stakeholders and people likely to be impacted by the services such as PwD, relatives, unpaid and paid carers, Lyrics and Lunch and The Brain Charity. Stage 2 will use quantitative data collected for WPs I and II on the effects of the intervention on quality of life, mood, carer burden, depressive symptoms and agitation levels and will map identified outcomes creating an impact map. This stage will identify what changes occurred and for whom. Stage 3 will provide a monetised value to each of the outcomes including those that do not have a price attached to them such as changes in quality of life. For outcome measures that do not have a clear monetary value, established SROI value banks such as Social Value UK will be used to identify suitable financial proxies⁸¹. Stage 4 will establish impact by accounting for attribution, deadweight, displacement and drop-off. These adjustments are made to ensure that social value is not overclaimed⁷⁹. Percentage values will be determined for attribution which considers changes or outcomes declared in the analysis that may not be due to just the singing intervention but may be due to other support services. A percentage value for the deadweight will be determined which considers how much change in the outcome values would have happened regardless of the singing intervention programme. Deadweight

will be calculated as a 10% reduction in overall value for every resource the participant identifies as an alternative to the singing support service, for example if participants attend other support groups. Stage 5 will calculate the SROI benefits by adding up and subtracting any negatives (attribution, distribution, drop-off and deadweight), and this will be compared to the investment cost of the service. This calculation will provide us with a SROI ratio which demonstrates the social value of the service in relation to the cost invested, for example an SROI ratio of £3.50: £1 means that for every pound invested there is £3.50 of social value created. Stage 6 will disseminate the findings with key stakeholders and recommendations created for future delivery of the service.”

VERSION 2 – REVIEW

REVIEWER	McCleery, Jenny Oxford Health NHS Foundation Trust, Mental Health
REVIEW RETURNED	11-Oct-2023

GENERAL COMMENTS	Abstract / Data and analysis: “Semi-structured interviews will be conducted with a subset of (n=40) people to further investigate (word missing from ms) the impact of singing services” There are a fair number of other typos throughout the manuscript. Strengths and limitations: As the editor commented, the lack of an independent control group is an important methodological limitation and it should feature in the bullet point summary of limitations of the study, not just the later discussion. Absence of blinding is not the only potential source of bias. In your response, you acknowledge that inability to follow-up beyond 1 week after the end of the time-limited interventions is another limitation. This should also be added to the bullet point list of study limitations and the later discussion. Introduction: In my opinion, the section on the rationale behind the interventions is now somewhat overdone and implies that you will be able to investigate the effects of different components within multi-domain interventions, which is not the case with this design. You should be careful not to imply otherwise (e.g. in statements such as “It is possible that and (sic) intervention combining both physical (dancing) and cognitive (singing) elements could lead to increased benefits for PwD and their carers.” Or “it may be the social aspects of the sessions which lead to the benefits for PwD and their carers.”) It is clear that these interventions were chosen because you wish to evaluate what the local charities provide, which is fine as an objective. Data analysis: You might wish to consider participant residence (community-dwelling or care home) as another covariate for the Brain Charity intervention. Impact on social isolation, for example, may differ considerably between these groups. I wish you luck with your study.
---

REVIEWER	Amano, Takashi Rutgers University Newark
REVIEW RETURNED	19-Oct-2023
GENERAL COMMENTS	Thanks to the authors for their sincere and thorough responses to my comments. All of my concerns have been addressed, and I believe the manuscript is of sufficient quality for publication.

VERSION 2 – AUTHOR RESPONSE

Reviewer 1:

Abstract / Data and analysis:

“Semi-structured interviews will be conducted with a subset of (n=40) people to further investigate (word missing from ms) the impact of singing services”

There are a fair number of other typos throughout the manuscript.

Response: Thank you for highlighting these typos, these have now been corrected in the manuscript. We have also checked the full manuscript for further typos which have now been corrected.

Strengths and limitations:

As the editor commented, the lack of an independent control group is an important methodological limitation and it should feature in the bullet point summary of limitations of the study, not just the later discussion. Absence of blinding is not the only potential source of bias.

In your response, you acknowledge that inability to follow-up beyond 1 week after the end of the time-limited interventions is another limitation. This should also be added to the bullet point list of study limitations and the later discussion.

Response: Thank you for this comment, we have added these limitations to the strengths and limitation section. The following sections have been added to the manuscript:

Strengths and Limitations section:

“A limitation of this study is the lack of an independent control group.”

“A further limitation is the inability to follow-up after the 12-week intervention period and therefore long-term effects of the interventions cannot be determined in this study.”

Discussion:

“Additionally, this study is unable to determine the long-term effects of the interventions and cannot determine whether any effects of the intervention persist or diminish after the 12-week intervention. Future research should include further follow-up assessments post-intervention to determine long-term effects.”

Introduction:

In my opinion, the section on the rationale behind the interventions is now somewhat overdone and implies that you will be able to investigate the effects of different components within multi-domain

interventions, which is not the case with this design. You should be careful not to imply otherwise (e.g. in statements such as “It is possible that and (sic) intervention combining both physical (dancing) and cognitive (singing) elements could lead to increased benefits for PwD and their carers.” Or “it may be the social aspects of the sessions which lead to the benefits for PwD and their carers.”) It is clear that these interventions were chosen because you wish to evaluate what the local charities provide, which is fine as an objective.

Response: Thank you for this comment, we have removed or revised these statements from the manuscript so to not imply that we are able to determine the effects of the individual components of the interventions. The following section has been revised:

“Existing literature often focuses on the singing element of these interventions [43], however there are many social aspects of the sessions which lead to the benefits for PwD and their carers [44,45] that have previously been overlooked.”

The following statement was removed from the manuscript:

“It is possible that singing interventions combining both physical (dancing) and cognitive (singing) elements could lead to increased benefits for PwD and their carers.”

Data analysis:

You might wish to consider participant residence (community-dwelling or care home) as another covariate for the Brain Charity intervention. Impact on social isolation, for example, may differ considerably between these groups.

Response: Thank you for this suggestion. We agree that this factor should be included as a covariate and we have added this to our analysis plan.

We would once again like to thank the reviewers for their further helpful comments aiding us in improving the manuscript.